# Diploid Nuclei Occur throughout the Life Cycles of Pucciniales Fungi

Pedro Talhinhas,[a] Rita Carvalho,[a] Sílvia Tavares,[b,c] Teresa Ribeiro,[a] Helena Azinheira,[a,c] Ana Paula Ramos,[a,d] Maria do Céu Silva,[a,c] Marta Monteiro,[e] João Loureiro,[f] Leonor Morais-Cecílio[a]

[a]LEAF-Linking Landscape, Environment, Agriculture and Food Research Centre and Terra Associated Laboratory, Instituto Superior de Agronomia, Universidade de Lisboa, Lisbon, Portugal

[b]Section for Plant and Soil Science, Department of Plant and Environmental Sciences, Faculty of Science, University of Copenhagen, Frederiksberg, Copenhagen, Denmark

[c]Centro de Investigação das Ferrugens do Cafeeiro, Instituto Superior de Agronomia, Universidade de Lisboa, Oeiras, Portugal

[d]LPVVA, Laboratório de Patologia Vegetal "Veríssimo de Almeida", Instituto Superior de Agronomia, Universidade de Lisboa, Lisbon, Portugal

[e]Instituto Gulbenkian de Ciência, Oeiras, Portugal

[f]CFE-Centre for Functional Ecology and Terra Associated Laboratory, Departamento de Ciências da Vida, Universidade de Coimbra, Coimbra, Portugal

**ABSTRACT** Within Eukaryotes, fungi are the typical representatives of haplontic life cycles. Basidiomycota fungi are dikaryotic in extensive parts of their life cycle, but diploid nuclei are known to form only in basidia. Among Basidiomycota, the Pucciniales are notorious for presenting the most complex life cycles, with high host specialization, and for their expanded genomes. Using cytogenomic (flow cytometry and cell sorting on propidium iodide-stained nuclei) and cytogenetic (FISH with rDNA probe) approaches, we report the widespread occurrence of replicating haploid and diploid nuclei (i.e., 1C, 2C and a small proportion of 4C nuclei) in diverse life cycle stages (pycnial, aecial, uredinial, and telial) of all 35 Pucciniales species analyzed, but not in sister taxa. These results suggest that the Pucciniales life cycle is distinct from any cycle known, i.e., neither haplontic, diplontic nor haplodiplontic, corroborating patchy and disregarded previous evidence. However, the biological basis and significance of this phenomenon remain undisclosed.

**IMPORTANCE** Within Eukaryotes, fungi are the typical representatives of haplontic life cycles, contrasting with plants and animals. As such, fungi thus contain haploid nuclei throughout their life cycles, with sexual reproduction generating a single diploid cell upon karyogamy that immediately undergoes meiosis, thus resuming the haploid cycle. In this work, using cytogenetic and cytogenomic tools, we demonstrate that a vast group of fungi presents diploid nuclei throughout their life cycles, along with haploid nuclei, and that both types of nuclei replicate. Moreover, haploid nuclei are absent from urediniospores. The phenomenon appears to be transversal to the organisms in the order Pucciniales (rust fungi) and it does not occur in neighboring taxa, but a biological explanation or function for it remains elusive.

**KEYWORDS** Basidiomycota, Pucciniales, rust fungi, nuclear cycle, haploid nuclei, diploid nuclei, FISH, flow cytometry

Address correspondence to Pedro Talhinhas, ptalhinhas@isa.ulisboa.pt.

The authors declare no conflict of interest.

Fungi are the typical representatives of haplontic life cycles, characterized by predominant haploid stages. In most cases, diploid nuclei occur only in a single cell (the basidium or the ascus, in Basidiomycota and Ascomycota, respectively) following karyogamy between haploid nuclei and immediately followed by meiosis. Basidiomycota and Ascomycota (i.e., the Dikarya) undergo plasmogamy before karyogamy, leading to a dikaryotic stage that is mostly short in the Ascomycota but that can be more prolonged in the Basidiomycota. Rust fungi (Basidiomycota, Pucciniales) are also reported to obey this general rule, with the basidium as the single diploid cell. The haploid cycle in rust fungi is divided into two stages, the

first comprising monokaryotic cells (from basidiospores to pycniospore formation), and the second, following plasmogamy, comprising dikaryotic cells (from pycniospore conjugation through aecial and uredinial stages to karyogamy occurring in teliospore leading to the diploid basidium).

The Pucciniales is a vast order in species number, but uniform in biological terms (1). Besides their complex life cycles and feeding through haustoria, rust fungi have several other unifying traits, including their biotrophic parasitic infection of plants, their (most frequently) specialization in single (or few) host species, their gene-for-gene interaction with their hosts, and their very large genomes. In fact, the Pucciniales contrast sharply with their close relatives in genome size, number of species and lifestyle. Furthermore, the Pucciniales are recognized as having the most complex life cycles in the fungal kingdom. This situation has shaped the evolution of these organisms through host specialization and, most likely, genome expansion (2). These traits are relevant for distinguishing the Pucciniales from their sister taxa, along with the absence of a cap of endoplasmic reticulum enclosing the spindle pole body discs during nuclear division (3). Furthermore, the Pucciniales, along with their sister orders (i.e., the class Pucciniomycetes), are notorious for undergoing mitosis and cell division without the formation of clamp connections (in most Basidiomycota, clamp connections ensure the maintenance of the heterokaryon upon cell division). Finally, Pucciniomycotina can be distinguished from the other subphyla in the Basidiomycota by cell wall sugar composition and traits in the septal pores and by the presence of disc-like spindle pole bodies (3, 4).

Intriguingly, our flow cytometric analyses aimed at genome size measurement indicate the occurrence of 1C, 2C, and a residual proportion of 4C nuclei across a range of species and families in the Pucciniales (5) in various life cycle stages, suggesting that diploid nuclei occur in stages that were supposed to be strictly haploid.

To describe this phenomenon and investigate its genetic and genomic basis, we have selected a set of Pucciniales species with contrasting life cycles (micro-to macrocyclic, including hemi- and demicyclic species) and analyzed their nuclei along such life cycles. Cytogenetic and cytogenomic analyses were performed using Fluorescent *In Situ* hybridization (FISH) and flow cytometry (FCM), respectively. To explore the phylogenetic limits of this phenomenon, we have extended the cytogenomic analysis to fungi in sister orders of the Pucciniales.

## RESULTS

**Ploidy level and nuclear DNA content determination.** Nuclei from 35 Pucciniales species were analyzed for ploidy level: two in the pycnial stage (0, following Cummins and Hiratsuka [1]), eight in the aecial stage (I), 21 in the uredinial stage (II) and nine in the telial stage (III). This included six microcyclic species, six hemicyclic species, two demicyclic species and 21 macrocyclic species. The demicyclic *Puccinia smyrnii* and the macrocyclic autoaecious *Puccinia obscura* were analyzed in stages I and III. *Gymnosporangium* spp. and *Puccinia phragmitis* were only analyzed on the aecial hosts (stages I or 0 and I, respectively). In contrast, *Melampsora pulcherrima* was analyzed both on the aecial (*Mercurialis annua*, stages 0 and I) and telial (*Populus alba*, stage II) hosts. *Puccinia coronata* on *Rhamnus alaternus* (stages 0 and I) and on *Avena sterilis* (stages II and III) was also analyzed but yielded poor results and was discarded from the study. A total of 40 samples were analyzed (Table 1).

For each Pucciniales species, flow cytometry results (Fig. 1) depict two (and in many cases three) nuclei populations with proportional nuclear DNA content: 1×, 2× (and 4×), indicating the presence of 1C, 2C and 4C fungal nuclei (e.g., Fig. 1A01). Additional nuclei populations in the histograms correspond to host plant nuclei (or to DNA standards; e.g., Fig. 1A01 and A05). In several situations, host plant nuclei are large enough to be out of the fluorescence range (e.g., Fig. 1A04). In a few situations, the host plant nuclei have roughly twice the DNA content of the 1C nuclei of the respective rust fungus, impairing the detection of a putative fungal 2C population (see Fig. 1A10 and A20 corresponding to *Uromyces appendiculatus* and *Tranzschelia discolor* fungi and host plants *Phaseolus vulgaris*

**TABLE 1** List of fungi analyzed in this study with reference to their phylogeny, host plant (for rust fungi), life cycle and stages examined, proportion of number of nuclei in 1C, 2C, and 4C populations, and fungus (and host) genome size

| Phylum | Subphylum | Class | Order | Family | Species (and isolate reference) | Host plant species | Rust life cycle[a] | Cycle stages examined[b] | Proportion of nuclei in each population (%) | | | Fungal genome size (1C, in Mbp) | Host plant genome size (2C, in Mbp) |
|---|---|---|---|---|---|---|---|---|---|---|---|---|---|
| | | | | | | | | | 1C | 2C | 4C | | |
| Basidiomycota | Pucciniomycotina | Pucciniomycetes | Pucciniales | Coleosporiaceae | Coleosporium tussilaginis f. sp. inulae | Dittrichia viscosa | Ma | II | 29.6 | 59.7 | 10.8 | 390.3 (5) | 224.6 (35) |
| | | | | | Coleosporium tussilaginis f. sp. tussilaginis | Tussilago farfara | Ma | II | 61.6 | 33.5 | 4.9 | 674.6[c] | 3263.4 (36) |
| | | | | Melampsoraceae | Melampsora euphorbiae | Ricinus communis | He | II | 73.4 | 24.8 | 1.8 | 332.8 (5) | 1020.0 (37) |
| | | | | | Melampsora pulcherrima | Mercurialis annua | Ma | 0 | 37.7 | 60.3 | 0.0 | 215.6 (38) | 1298.5 (39) |
| | | | | | | Populus alba | | | 14.3 | 81.7 | 4.0 | | 1019.2 (40) |
| | | | | Phakopsoraceae | Phakopsora pachyrhizi (UoH-Thai1) | Glycine max | He | I | 27.4 | 52.4 | 20.2 | 720.2 (5) | 2214.8 (41) |
| | | | | Pucciniaceae | Gymnosporangium confusum | Crataegus monogyna | Ma | I | 42.5 | 50.9 | 6.6 | 893.3 (5) | 1489.6 (42) |
| | | | | | Gymnosporangium sp. | Cydonia oblonga | He | I | 65.3 | 34.7 | 0.0 | 434.3[c] | 1470.0 (43) |
| | | | | | Miyagia pseudosphaeria | Sonchus oleraceus | Mi | III | 68.5 | 31.5 | 0.0 | 659.0[c] | 3136.0[d] |
| | | | | | Puccinia arenariae | Silene latifolia | Mi | III | 41.7 | 58.3 | 0.0 | 182.1 (38) | 5292.0 (44) |
| | | | | | Puccinia buxi | Buxus sempervirens | Mi | III | 65.6 | 34.4 | 0.0 | 657.1 (5) | 1411.0 (45) |
| | | | | | Puccinia difformis | Galium aparine | De | I | 38.9 | 59.4 | 1.7 | 284.1[c] | 2842.0 (46) |
| | | | | | Puccinia galactitis | Galactites tomentosa | Mi | III | 46.6 | 26.2 | 27.1 | 122.3[c] | 1960.0 (35) |
| | | | | | Puccinia hemerocallidis | Hemerocallis sp. | Ma | II | 66.2 | 33.8 | 0.0 | 345.0 (29) | 9486.8[e] |
| | | | | | Puccinia hordei | Hordeum vulgare | Ma | II | 26.3 | 62.4 | 11.3 | 237.2 (5) | 10878.0 (47) |
| | | | | | Puccinia malvacearum | Lavatera cretica | Mi | III | 76.4 | 23.6 | 0.0 | 177.8 (5) | n.a.[e] |
| | | | | | Puccinia mesnieriana | Rhamnus alaternus | Mi | III | 33.3 | 59.3 | 7.4 | 67.0[c] | 663.7 (29) |
| | | | | | Puccinia obscura | Bellis perennis | Ma | III | 35.7 | 64.3 | 0.0 | 213.4[c] | 3090.0 (48) |
| | | | | | Puccinia oxalidis | Oxalis articulata | Ma | III | 75.2 | 24.8 | 0.0 | 354.9 (5) | 888.2 (49) |
| | | | | | Puccinia pelargonii-zonalis | Pelargonium x hortorum | He | II | 63.7 | 32.2 | 4.1 | 183.6 (5) | 1754.2 (50) |
| | | | | | Puccinia phragmitis | Rumex sp. | Ma | 0 | 54.1 | 45.9 | 0.0 | 168.1[c] | indet.[e] |
| | | | | | Puccinia pimpinellae | Pimpinella villosa | Ma | III | 60.6 | 34.5 | 4.9 | 321.6 (38) | n.a.[e] |
| | | | | | Puccinia poarum | Senecio vulgaris | Ma | I | 54.4 | 45.6 | 0.0 | 86.6[c] | 3136.0 (51) |
| | | | | | Puccinia porri | Allium ampeloprasum subsp. porrum | Ma | 0 | 36.0 | 64.0 | 0.0 | 351.7 (5) | 58702[d] |
| | | | | | Puccinia smyrnii | Smyrnium olusatrum | De | I | 48.6 | 45.5 | 5.9 | 258.9 (5) | 6584.2[c] |
| | | | | | Puccinia verruca | Centaurea pullata | Mi | III | 33.4 | 60.3 | 6.3 | 143.3[c] | n.a.[e] |
| | | | | | Puccinia vinca | Vinca difformis | Ma | III | 20.1 | 69.6 | 10.2 | 566.4 (38) | 2322.0 (52) |
| | | | | | Uromyces appendiculatus (UoH-SWBR) | Phaseolus vulgaris | Ma | II | 29.0 | 64.8 | 6.2 | 679.4 (5) | 1176.0[d] |
| | | | | | Uromyces bidentis | Bidens pilosa | Ma | II | 53.6 | 46.4 | 0.0 | 2489.0 (53) | 3332.0[d] |
| | | | | | Uromyces dianthi | Dianthus caryophyllus | Ma | II | 70.0 | 30.0 | 0.0 | 379.4 (5) | 2548.0 (54) |
| | | | | | Uromyces fabae (UoH-race 12) | Vicia faba | Ma | II | 22.8 | 77.2 | 0.0 | 415.2 (5) | 26166 (47) |
| | | | | | Uromyces striatus (UoH-Us) | Medicago arabica | He | II | 27.4 | 67.7 | 4.9 | 282.8 (5) | 1195.6 (55) |
| | | | | | Uromyces transversalis | Gladiolus sp. | Ma | II | 42.3 | 57.7 | 0.0 | 302.6 (5) | indet.[e] |
| | | | | Pucciniastraceae | Pucciniastrum epilobii | Fuchsia sp. | Ma | II | 49.2 | 41.2 | 9.6 | | indet.[e] |
| | | | | Raveneliaceae | Tranzschelia discolor | Prunus dulcis | Ma | II | 28.9 | 71.1 | 0.0 | | 646.8 (56) |
| | | | | Zaghouaniaceae | Hemileia vastatrix (CIFC178a) | Coffea arabica | He | II | 36.5 | 53.6 | 9.9 | 796.8 (5) | 2352.0 (57) |
| | | | | | | | | | 26.2 | 60.4 | 13.5 | | |
| | | | | | | | | | 32.5 | 57.2 | 10.4 | | |
| | | | Helicobasidiales | Helicobasidiaceae | Tuberculina maxima (CBS136.66) | | | | 64.1 | 35.9 | 0.0 | 35.16[c] | |
| | | | Platygloeales | Platygloeaceae | Herpobasidium deformans (CBS197.56) | | | | 44.7 | 48.8 | 6.5 | 58.45[c] | |
| | | | Pachnocybales | Pachnocybaceae | Pachnocybe ferruginea (CBS426.88) | | | | 80.2 | 19.8 | 0.0 | 45.88[c] | |
| | | | Septobasidiales | Septobasidiaceae | Septobasidium carestianum (CBS101450) | | | | 100.0 | 0.0 | 0.0 | ND | |
| | | Microbotryomycetes | Microbotryales | Microbotryaceae | Microbotryum lychnidis-dioicae (PYCC4281) | | | | 83.8 | 16.2 | 0.0 | 29.9 (58) | |
| | | | Sporidiobolales | Sporidiobolaceae | Rhodotorula babjevae (CBS7809) | | | | 100.0 | 0.0 | 0.0 | ND | |
| | Agaricomycotina | Agaricomycetes | Hymenochaetales | Hymenochaetaceae | Inonotus hispidus (LPV629) | | | | 86.2 | 13.8 | 0.0 | 41.2 (30) | |
| | | | Cantharellales | Ceratobasidiaceae | Rhizoctonia sp. (LPV630) | | | | 91.6 | 8.4 | 0.0 | | |
| Ascomycota | Pezizomycotina | Sordariomycetes | Glomerellales | Glomerellaceae | Colletotrichum acutatum (PT812) | | | | 80.9 | 19.1 | 0.0 | 67.2 (30) | |
| | | | Xylariales | Apiosporaceae | Nigrospora sphaerica | | | | 88.2 | 11.8 | 0.0 | 64.8[c] | |
| | | Dothideomycetes | incertae sedis | Gloniaceae | Cenococcum geophilum (844.1) | | | | 97.2 | 2.8 | 0.0 | 203.0 (26) | |
| | Saccharomycotina | Saccharomycetes | Saccharomycetales | Saccharomycetaceae | Saccharomyces cerevisiae (CP1969) | | | | 84.0 | 16.0 | 0.0 | 23.0 (30) | |

[a]Rust life cycles: Ma—macrocyclic; Mi—microcyclic; He—hemicyclic; De—demicyclic.

[b]Cycle stages examined: 0, pycnial stage; I, aecial stage; II, uredinial stage; III, telial stage; IV, basidia/basidiospores.

[c]The DNA standards used in this study for genome sizes measurement were: Cg—Cenococcum geophilum, 0.208 pg DNA/1C (26, 30) (used for Tuberculina maxima, Herpobasidium deformans, Pachnocybe ferruginea and Nigrospora sphaerica); Ca—Colletotrichum acutatum, 0.0689 pg DNA/1C (30) (used for Rhizoctonia sp.); Rs—Raphanus sativus, 1.11 pg DNA/2C (28) (used for Coleosporium tussilaginis f. sp. tussilaginis, Puccinia difformis, P. obscura and P. poarum); Ra—Rhamnus alaternus, 0.680 pg DNA/2C (29) (used for Puccinia galactitis, P. mesnieriana, P. phragmitis and P. verruca); Sl—Solanum lycopersicum, 1.96 pg DNA/2C (28) (used for Gymnosporangium sp., Miyagia pseudosphaeria and Puccinia smyrnii); Vf—Vicia faba, 26.65 pg DNA/2C (28) (used for Hemerocallis sp.).

[d]Unpublished values available at the Plant DNA C-values Database, Royal Botanic Gardens, Kew and attributed to: Band, S. R. (personal communication. 1984) for Sonchus oleraceus; Ohri, D., Fritsch, R., and Hanelt, P. (personal communication. 1996) for Allium ampeloprasum subsp. porrum; Kenton, A. Y., and Owens, S. J. (personal communication. 1988) for Phaseolus vulgaris; the Jodrell Laboratory (Royal Botanic Gardens, Kew) for Bidens pilosa.

[e]indet., indeterminate; n.a., not available; ND, not determined.

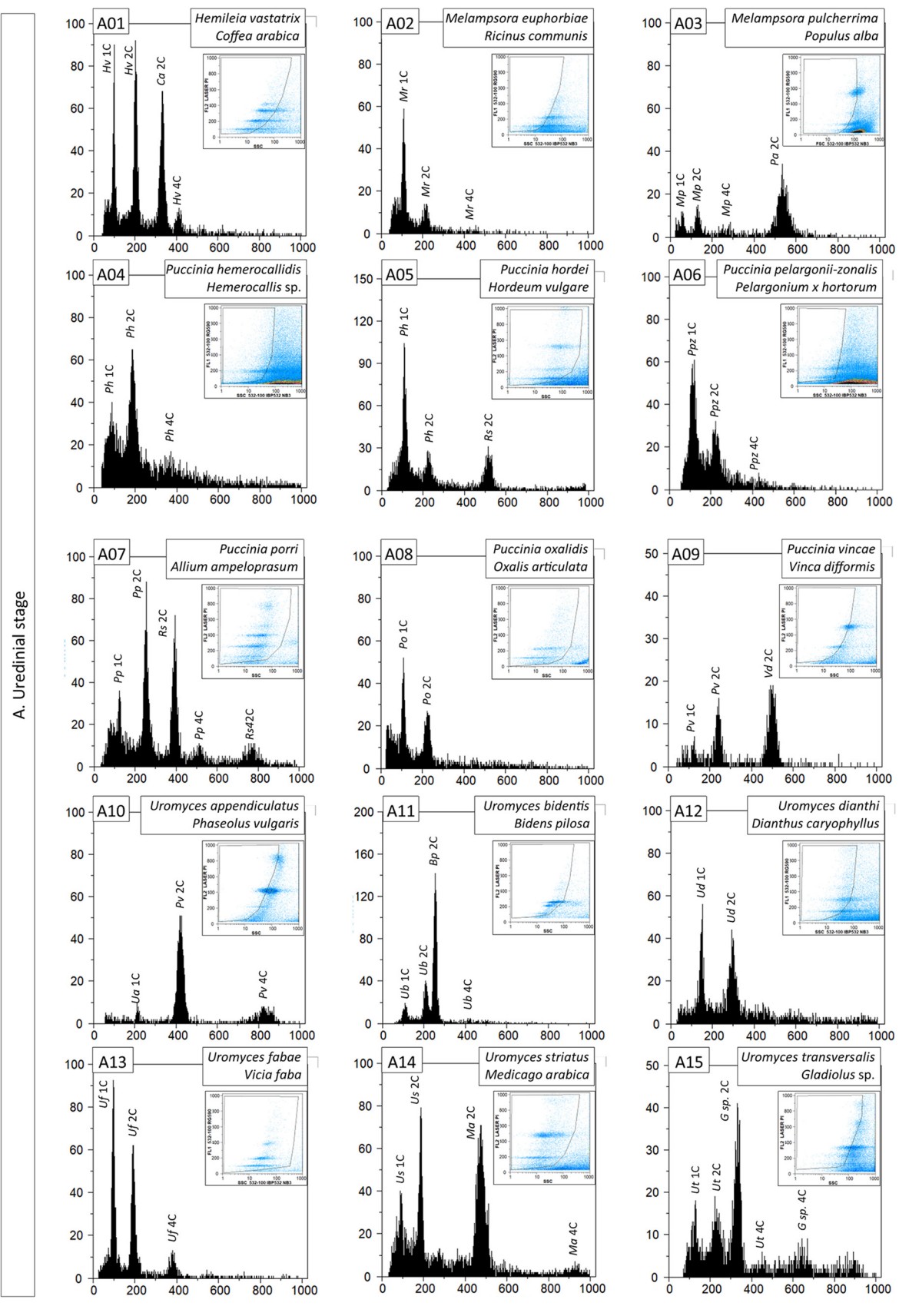

**FIG 1** (Continued)

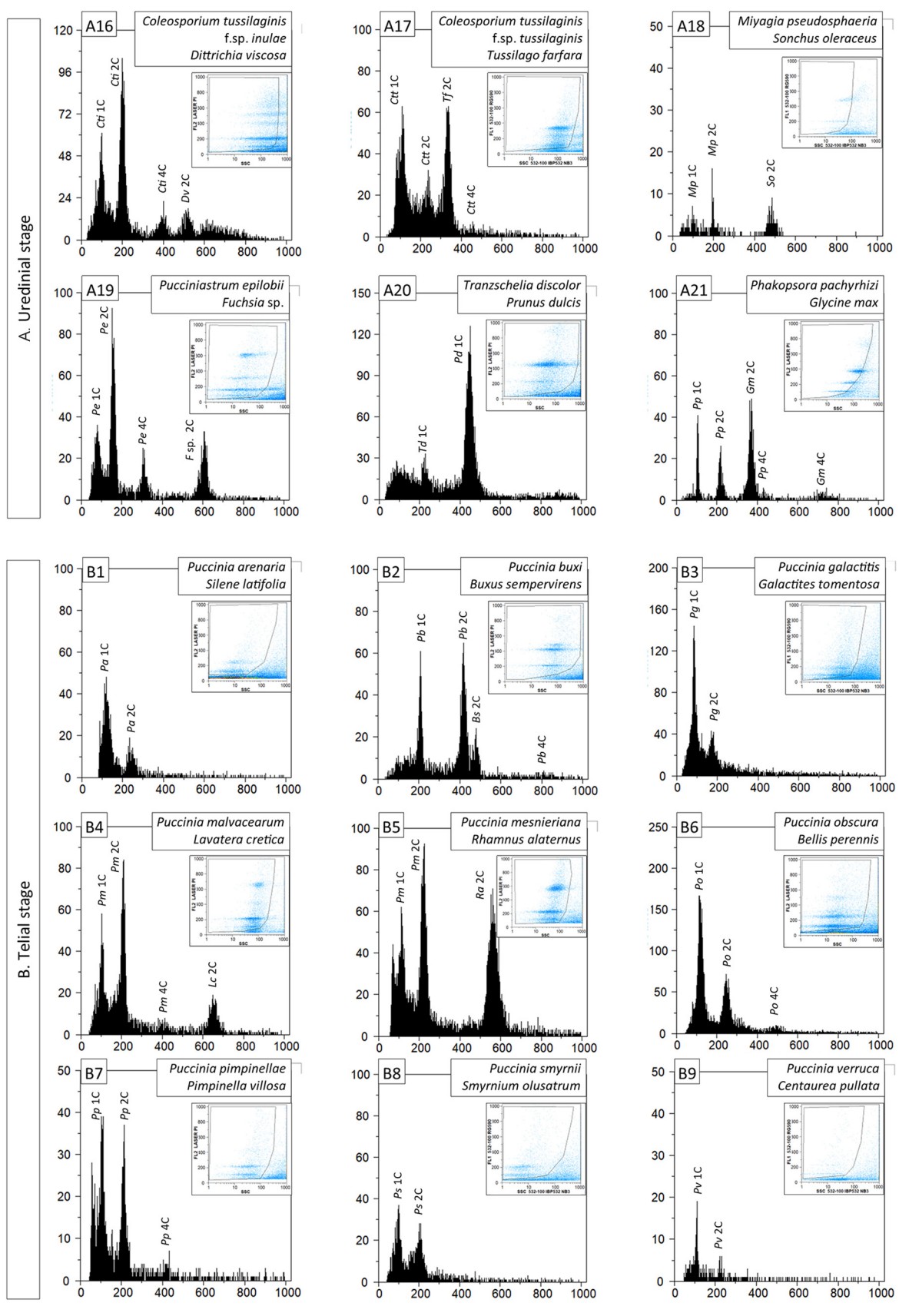

**FIG 1** (Continued)

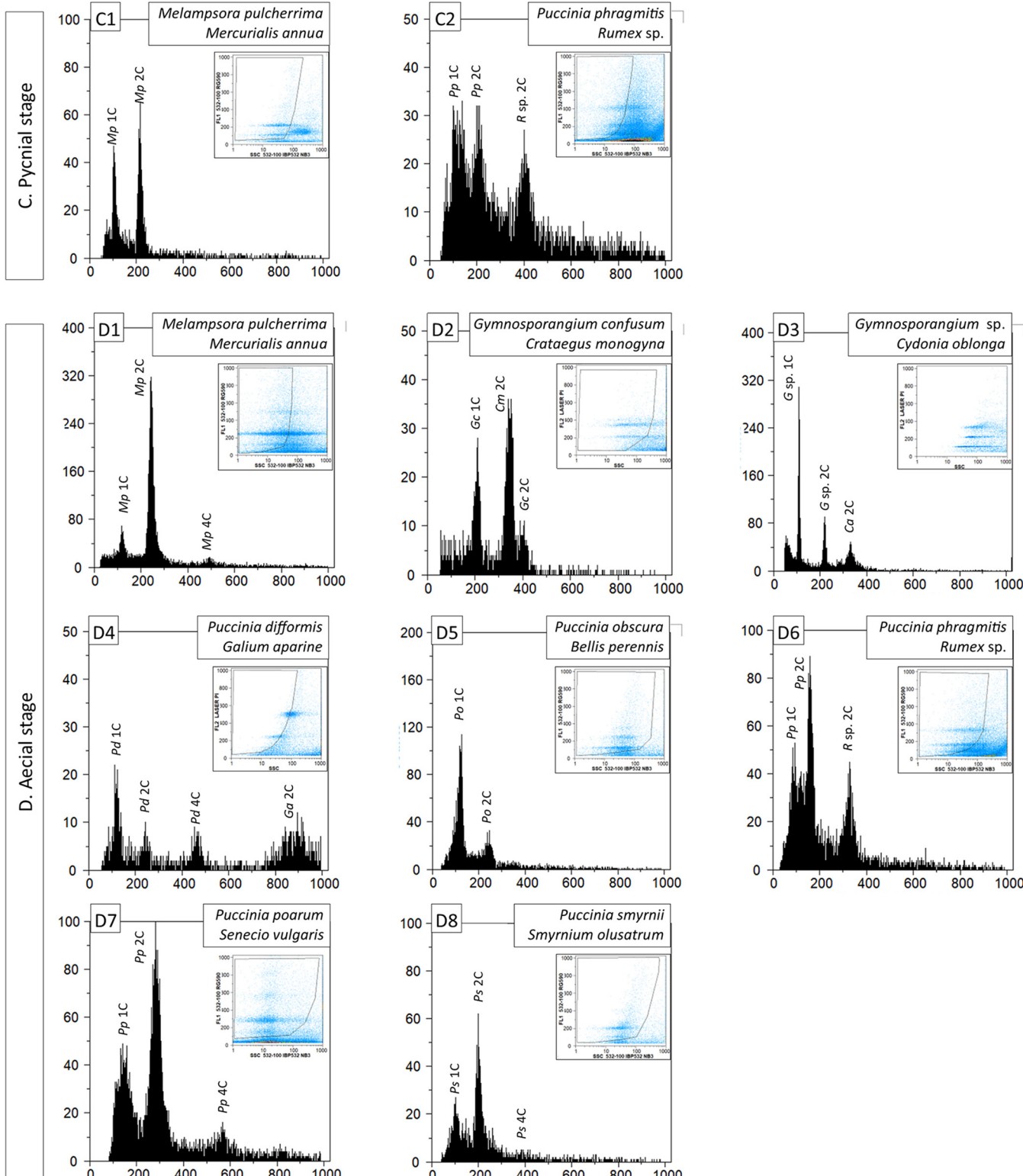

**FIG 1** Flow cytometric histograms (number of nuclei depicted in vertical axes) of relative fluorescence intensities (horizontal axes) of propidium iodide-stained nuclei simultaneously isolated from the rust fungus and its host plant (scientific names given in each panel), according to the rust life cycle stage sampled (plates A to D), as detailed in Table 1. In each histogram, peaks are identified for ploidy (1C, 2C, and 4C) and with the organism's acronym (Ra and Rs refer to the DNA standards *Raphanus sativus* and *Rhamnus alaternus*, respectively); in each histogram, the inset graphic represents the gating made in the dot-plot of side scatter (SSC) versus relative fluorescence (FL) to exclude as much as possible partial nuclei and other types of debris.

**TABLE 2** Proportion (%) of 1C, 2C and 4C nuclei isolated from samples of Pucciniales fungi obtained at diverse life cycle stages (mean ± standard deviation; proportions for each species are given in Table 1)

| Life cycle stage | Ploidy level | | | Nr. of samples |
|---|---|---|---|---|
| | 1C | 2C | 4C | |
| Pycnial (0) | 47.1 (±10.4) | 52.9 (±10.4) | 0.0 (±0.0) | 2 |
| Aecial (I) | 46.0 (±21.7) | 48.5 (±21.7) | 5.5 (±9.2) | 8 |
| Uredinial (II) | 41.0 (±17.2) | 52.7 (±15.3) | 6.3 (±5.8) | 19 |
| Telial (III) | 52.9 (±14.3) | 45.0 (±13.3) | 2.1 (±2.9) | 9 |
| All | 45.2 (±17.4) | 50.0 (±16.0) | 4.8 (±6.2) | 38 |

and *Prunus dulcis*, respectively; a similar situation was observed for *T. discolor* on *Prunus persica*; data not shown). However, the analysis of nuclei extracted from *U. appendiculatus* resting and germinating urediniospores showed the presence of nuclei with twice the DNA content of those obtained from infected leaves (Fig. 4G to I). Thus, all rust samples analyzed, irrespective of the life cycle stage sampled, exhibited 1C and 2C nuclei populations with an equivalent number of nuclei (Table 1). Additionally, a smaller proportion of 4C nuclei were detected in 20 out of 40 samples analyzed. In eight samples, host plant nuclei have relative fluorescence values similar to those of putative 4C populations, whereas in the remaining 12 samples the 4C population was not detected. The 4C population was not detected in any of the two samples collected in the pycnial stage, but was present in the remaining sampled stages. The proportion of nuclei in the 1C, 2C and 4C populations is 45%, 50% and 5%, respectively. These frequencies are steady, with few variations, regardless of the life cycle stage (Table 2).

In this study, the genome size of 10 Pucciniales species was newly determined (Table 1), ranging between 67.0 Mbp for *Puccinia mesnieriana* (on *Rhamnus alaternus*) and 674.9 Mbp for *Coleosporium tussilaginis* f. sp. *tussilaginis* (on *Tussilago farfara*). With a genome size of 659.0 Mbp estimated for *Miyagia pseudosphaeria*, this is the first genome size record for the genus *Miyagia*. Among the 10 species newly analyzed, six were found on Asteraceae plants, being noticeable the small genome sizes of *P. poarum* on *Senecio vulgaris* (86.6 Mbp), of *P. galactitis* on *Galactites tomentosa* (122.3 Mbp), and of *P. verruca* on *Centaurea pullata* (143.3 Mbp).

Four fungal species representing sister orders to the Pucciniales (in the class Pucciniomycetes) were analyzed for ploidy level and DNA content determination, along with two species in the class Microbotryomycetes, sister to the Pucciniomycetes in the subphylum Pucciniomycotina (Fig. 2). Genome sizes of any of these six organisms are much smaller than in Pucciniales fungi, ranging between 29.9 Mbp (for *Microbotryum lychnidis-dioicae*) and 58.5 Mbp (for *Herpobasidium deformans*). In addition, a single nuclei population (1C) was detected in *Rhodotorula babjevae* (Fig. 2B2), whereas for *Microbotryum lychnidis-dioicae* a residual 2C nuclei population was detected along with the dominant 1C population (Fig. 2B1). Among the Pucciniomycetes, contrasting situations were observed: only one nuclei population (1C) was observed for *Septobasidium carestianum* (genome size not determined; Fig. 2A4); a dominant 1C population and a residual 2C population were observed in *Pachnocybe ferruginea* (genome size 45.9 Mbp; Fig. 2A3); for *Tuberculina maxima* (genome size 35.2 Mbp), 1C, and 2C nuclei populations were observed, representing, respectively, 64% and 36% of sampled nuclei (Fig. 2A1); in *Herpobasidium deformans* (58.5 Mbp), 1C, 2C, and 4C populations were observed (Fig. 2A2), representing, respectively, 45%, 49%, and 6% of sampled nuclei, thus following the pattern described for Pucciniales (Table 2).

Another two Basidiomycota (*Innonotus hispidus* and *Rhizoctonia* sp.; Fig. 2C) and four Ascomycota (*Cenococcum geophilum*, *Colletotrichum acutatum*, *Nigrospora sphaerica* and *Saccharomyces cerevisiae*; Fig. 2D) fungi were analyzed, in all cases showing prevalent 1C nuclei populations (88% of nuclei in average) and residual 2C nuclei populations (12%). Similar proportions of 1C and 2C nuclei were obtained upon analysis of

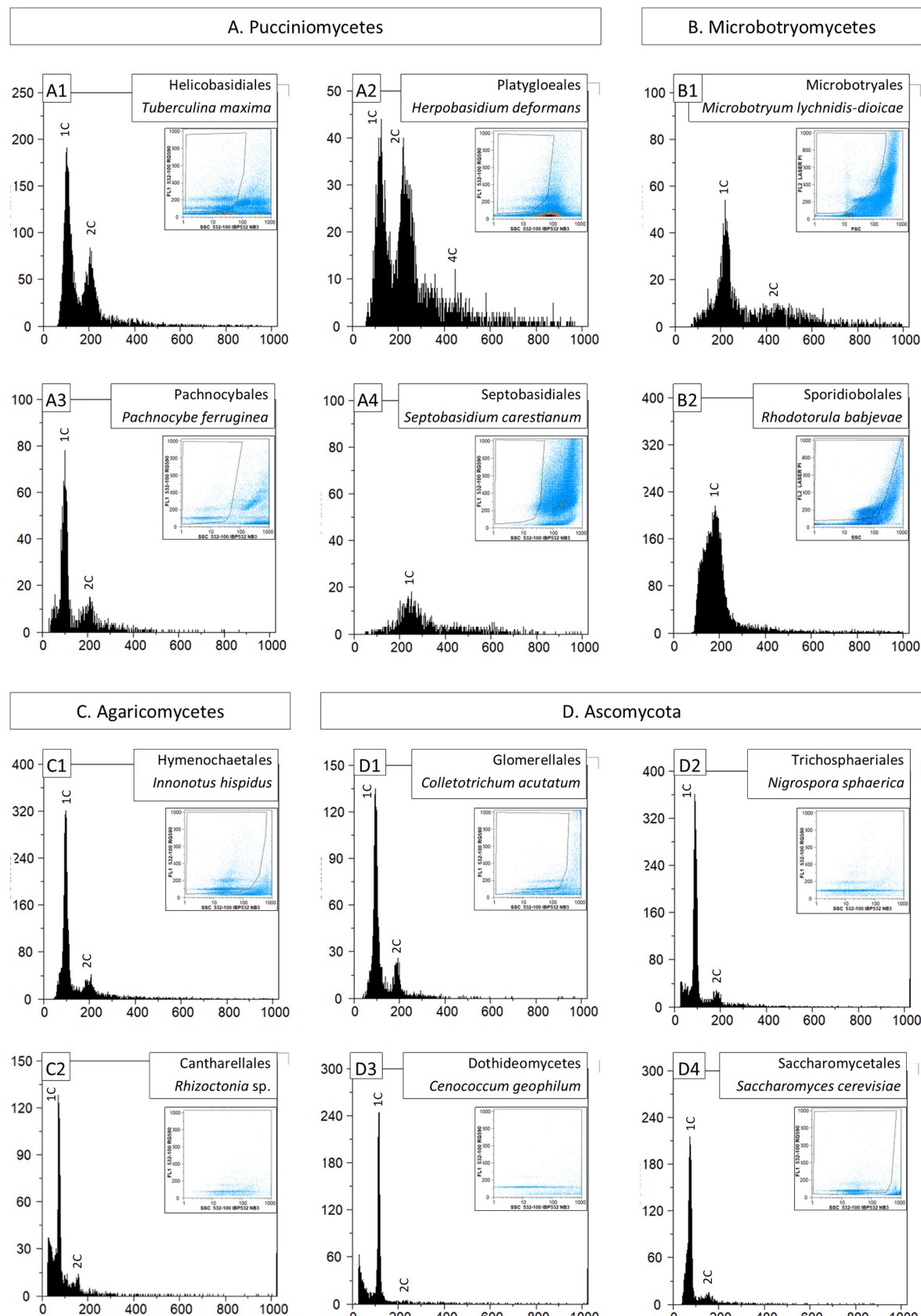

**FIG 2** Flow cytometric histograms (number of nuclei depicted in vertical axes) of relative fluorescence intensities (horizontal axes) of propidium iodide-stained nuclei isolated from non-Pucciniales fungi: Pucciniomycetes fungi (panel A); Microbotryomycetes fungi (panel B);

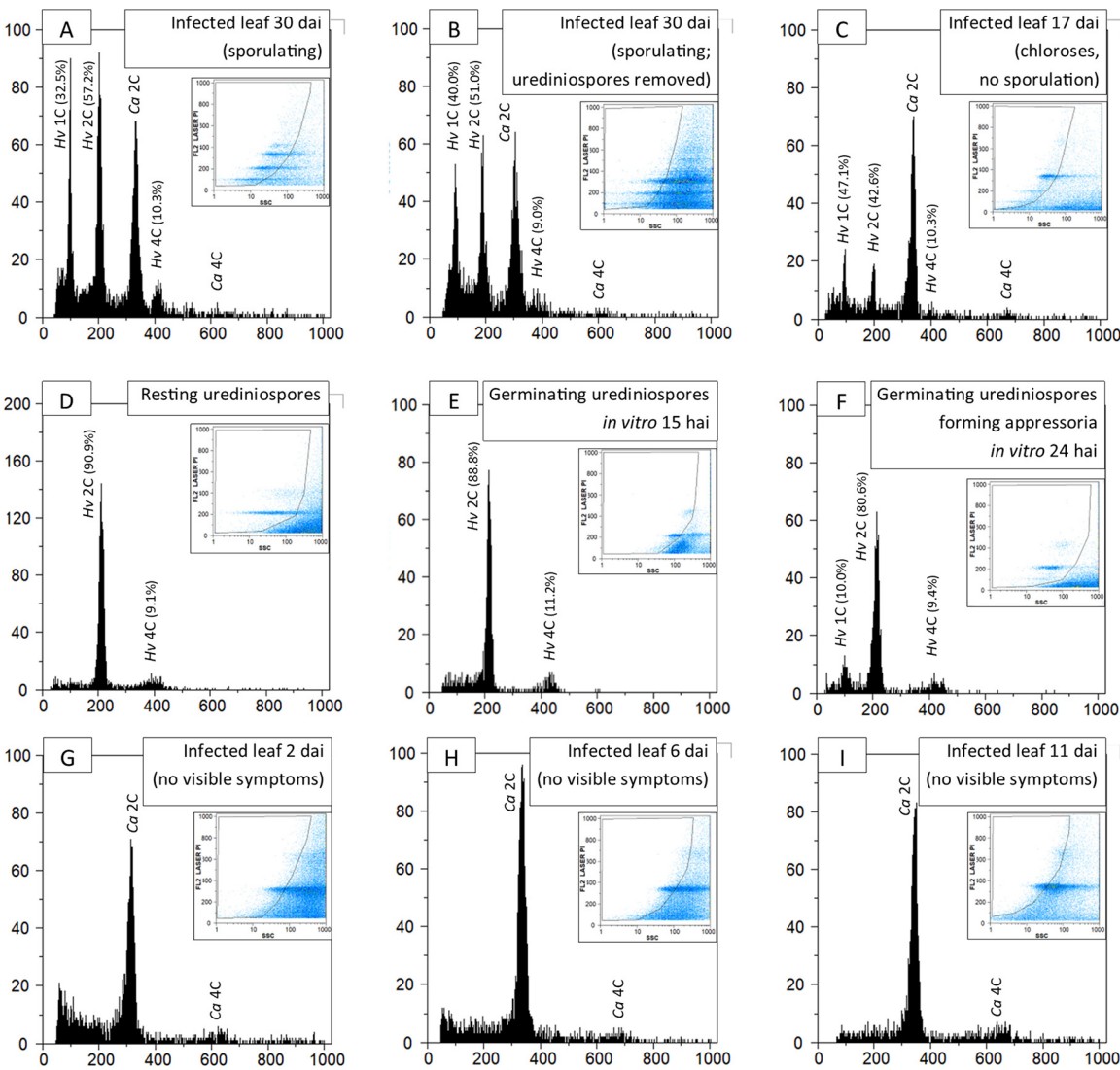

**FIG 3** Flow cytometric histograms (number of nuclei depicted in vertical axes) of relative fluorescence intensities (horizontal axes) of propidium iodide-stained nuclei simultaneously isolated from *Hemileia vastatrix* and *Coffea arabica* (or from *H. vastatrix* only; panels D to F) along the uredinial cycle stages of the fungus. In each histogram, peaks are identified with the organism's acronym and for ploidy (1C, 2C, and 4C), with percentages representing the relative proportion of fungal nuclei in each population. In each histogram, the inset graphic represents the gating made in the dot-plot of side scatter (SSC) versus relative fluorescence (FL) to exclude as much as possible partial nuclei and other types of debris. hai, hours after inoculation. dai, days after inoculation.

leaf samples of tomato, pea, soybean, fava bean and Arabica coffee (92% and 8%, respectively; data not shown).

The *Hemileia vastatrix* uredinial cycle was analyzed in detail to assess the presence of the nuclei populations in diverse stages (Fig. 3). In late, sporulating infections, 1C, 2C and a residual 4C nuclei populations were detected (Fig. 3A). *Hemileia vastatrix* produces suprastomatal, determinate sori (6), enabling a thorough removal of urediniospores (gently rinsed in 0.1% Tween 20 with a paint brush). Such late infections with mature spores removed (Fig. 3B) still revealed the presence of the three nuclei population types. However, the proportion of 2C nuclei (51.0%) was lower than in the sample containing spores (57.2%). Detailed dissection of *in planta* fungal structures using Laser Capture

**FIG 2** Legend (Continued)
Agaricomycetes fungi (panel C); and Ascomycota fungi (panel D). Details given in Table 1. In each histogram, peaks are identified for ploidy (1C, 2C, and 4C); in each histogram, the inset graphic represents the gating made in the dot-plot of side scatter (SSC) versus relative fluorescence (FL) to exclude as much as possible partial nuclei and other types of debris.

Microdissection was attempted but yielded no reliable FCM results. Similarly, no reliable results were obtained in isolated haustoria. In an earlier infection stage, prior to sporulation (17 days after inoculation, at the onset of the development of visible chloroses; Fig. 3C), fungal nuclei were much scarcer (for each fungal nucleus there are 2.4 plant nuclei, as opposed to the late infection sample, with 0.54 plant nuclei per each fungal nucleus). However, the three fungal nuclei populations are still detectable, with the 2C population representing 42.6% of all fungal nuclei. No fungal nuclei were detected in earlier infection stages (Fig. 3G to I), despite multiple attempts and long-running FCM samples. In resting urediniospores the 2C and 4C nuclei populations are present, but the 1C population is absent (Fig. 3D), similarly to germinating urediniospores (Fig. 3E). Concomitantly with appressoria formation (from 17 h after inoculation onwards; *in vitro*), the 1C population reappears, representing 4% of all fungal nuclei 17 h after inoculation, 5.4% 19 h after inoculation and 10.0% 24 h after inoculation (Fig. 3F; 25% appressoria formation rate, as related to the number of germinated urediniospores). Urediniospore germination and appressoria formation was also induced *in planta* and such samples were subjected to FCM analysis, but no fungal nuclei were detected.

Results similar to those reported for *H. vastatrix* were registered along the *in vitro* germination of *Uromyces striatus*, *U. fabae* and *U. appendiculatus* (Fig. 4): the 1C nuclei populations are absent from resting urediniospores and they reappear concomitantly with appressoria differentiation.

**Cytogenetic analysis.** FISH using 35S rDNA probes showed four patterns on rust fungi nuclei: a single dot, one paired-dot, two distinct single dots and two distinct paired-dots (Fig. 5). Whereas in the sample collected at the pycnial stage (Fig. 5, *Melampsora pulcherrima* (0) panels) only a single dot and one paired-dot nucleus were detected, two single dots and two paired-dots nuclei were additionally detected in samples collected at the aecial, uredinial and telial stages either in nuclei sorted by FACS or in chopped samples (Fig. 5 for fungi at life cycle stages I, II and III). The analysis of 1C, 2C and 4C nuclei collected by FACS (Fig. 5, FACS panels) enables associating the single-dot nuclei to the 1C population (1n1C), the one paired-dot and the two single dots nuclei to the 2C nuclei (1n2C and 2n2C, respectively) and the two paired-dots nuclei to the 4C population (2n4C). The relative proportion of nuclei in the four stages reported was estimated (Table 3) and follows the proportion of 1C, 2C and 4C nuclei populations identified by FCM for the same fungal species.

FISH was attempted in intact spores, appressoria and haustoria but with little success. While DAPI readily entered these structures and stained the nuclei, the rDNA probe could not be detected.

## DISCUSSION

In this work we have shown that 1C, 2C and a residual proportion of 4C nuclei occur across a range of species and families in the Pucciniales order and across their life cycles, both by FCM and by FISH. The identification by FCM of large proportions of 2C nuclei suggested that these are not merely replicating haploid nuclei. FISH with rDNA probes further showed that these 2C nuclei comprise both haploid replicating nuclei (1n2C) and diploid nonreplicating nuclei (2n2C). Moreover, the detection of 4C nuclei by FCM suggests that these diploid (2n2C) nuclei replicate, which was confirmed by FISH, depicting 2n4C nuclei. Replicating diploid nuclei are found in the aecial, uredinial and telial stages. Diploid nuclei may be absent in pycnial structures, although the number of samples analyzed in that stage is small. Moreover, haploid nuclei are absent in resting urediniospores, where only 2C and 4C nuclei are found, and they reappear during germination and appressoria formation.

It seems clear that this phenomenon is transversal to the whole Order, even if the number of species analyzed is a small fraction of all species in the Pucciniales. All the samples are consistent with these results. In fact, none of the rust species analyzed presented results contradictory to these findings. The phenomenon is mainly restricted to the Pucciniales, although similar results were obtained for the neighbor taxon *Herpobasidium deformans*

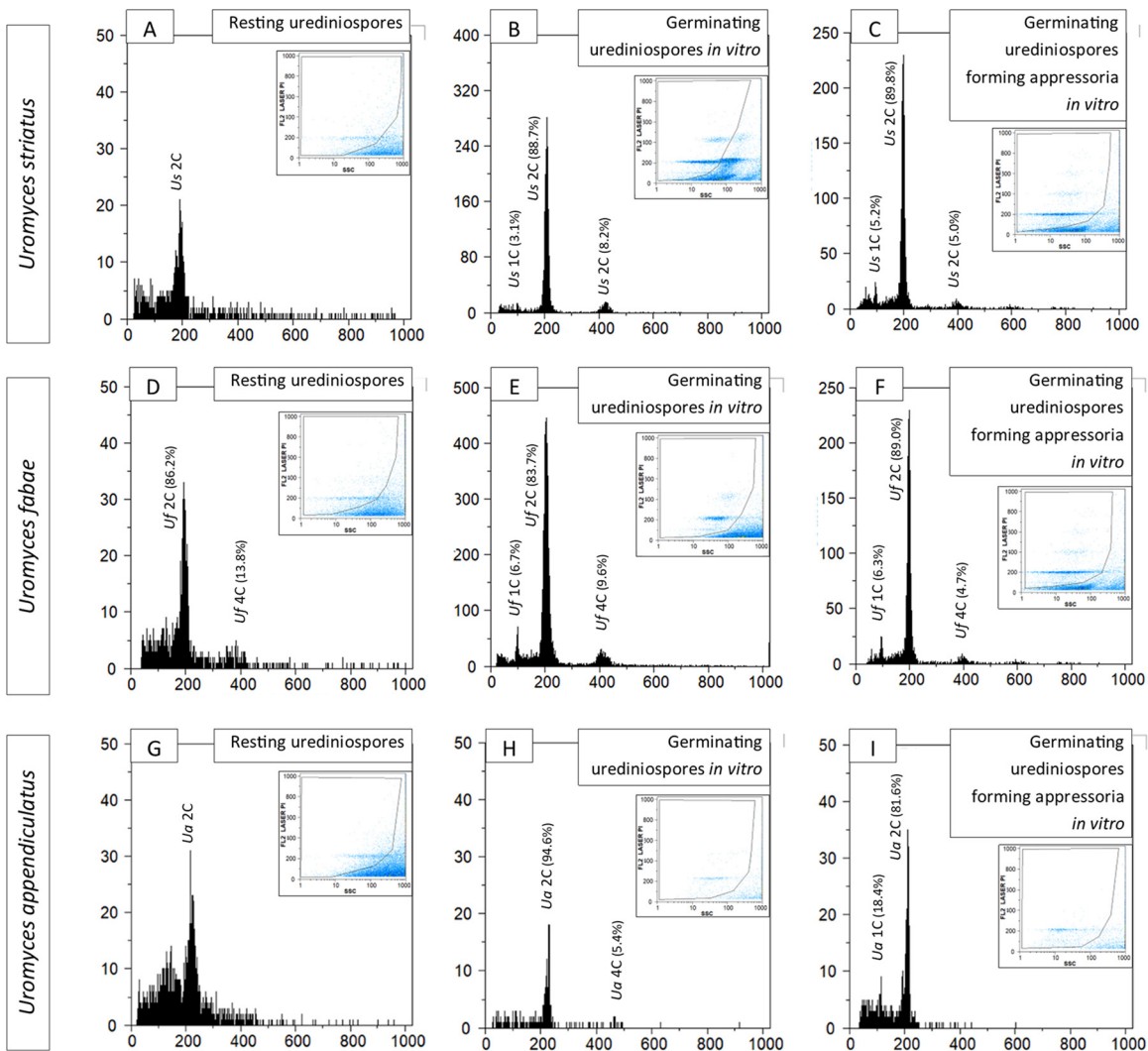

**FIG 4** Flow cytometric histograms (number of nuclei depicted in vertical axes) of relative fluorescence intensities (horizontal axes) of propidium iodide-stained nuclei simultaneously isolated from Pucciniales fungi along *in vitro* urediniospore germination and appressoria formation. In each histogram, peaks are identified with the organism's acronym and for ploidy (1C, 2C, and 4C), with percentages representing the relative proportion of fungal nuclei in each population. In each histogram, the inset graphic represents the gating made in the dot-plot of side scatter (SSC) versus relative fluorescence (FL) to exclude as much as possible partial nuclei and other types of debris.

(Platygoleales). Although many phylogenetically scattered exceptions to the general "long haploid/single-celled diploid" rule are known, and some fungi are actually stable diploids (e.g., [7]), our results consistently indicate an "exception" at the Order rank, challenging well-established text-book mycological principles stating that karyogamy is immediately followed by meiosis, and thus diploid nuclei should occur only in rust fungi basidia. Genomic data suggest that the early evolution of fungi may have been characterized by diplontic life cycles, reflected in similarities between protists and zoosporic fungi (8), a character that may have been retained in the Dikarya by rust fungi, according to our results, and presumably also in the heterokaryons of Arbuscular Mycorrhizal Fungi (9). Interestingly, the examination of ancient publications suggests that the "long haploid/single-celled diploid" rule has been transposed to the Pucciniales by inference from other fungi, rather than experimentally demonstrated in organisms of this order. Any textbook on mycology states this rule, but to the best of our knowledge, none gives cites experimental work showing this.

In fact, a few examples in the Pucciniales had been noted in the past, but no generalization to the Order level was attempted. In the 1960s, microscopic examination of the coffee leaf rust (*Hemileia vastatrix*) uredinial infection cycle suggested that karyogamy would occur in sporogenic hyphae in uredinia and that meiosis would occur throughout

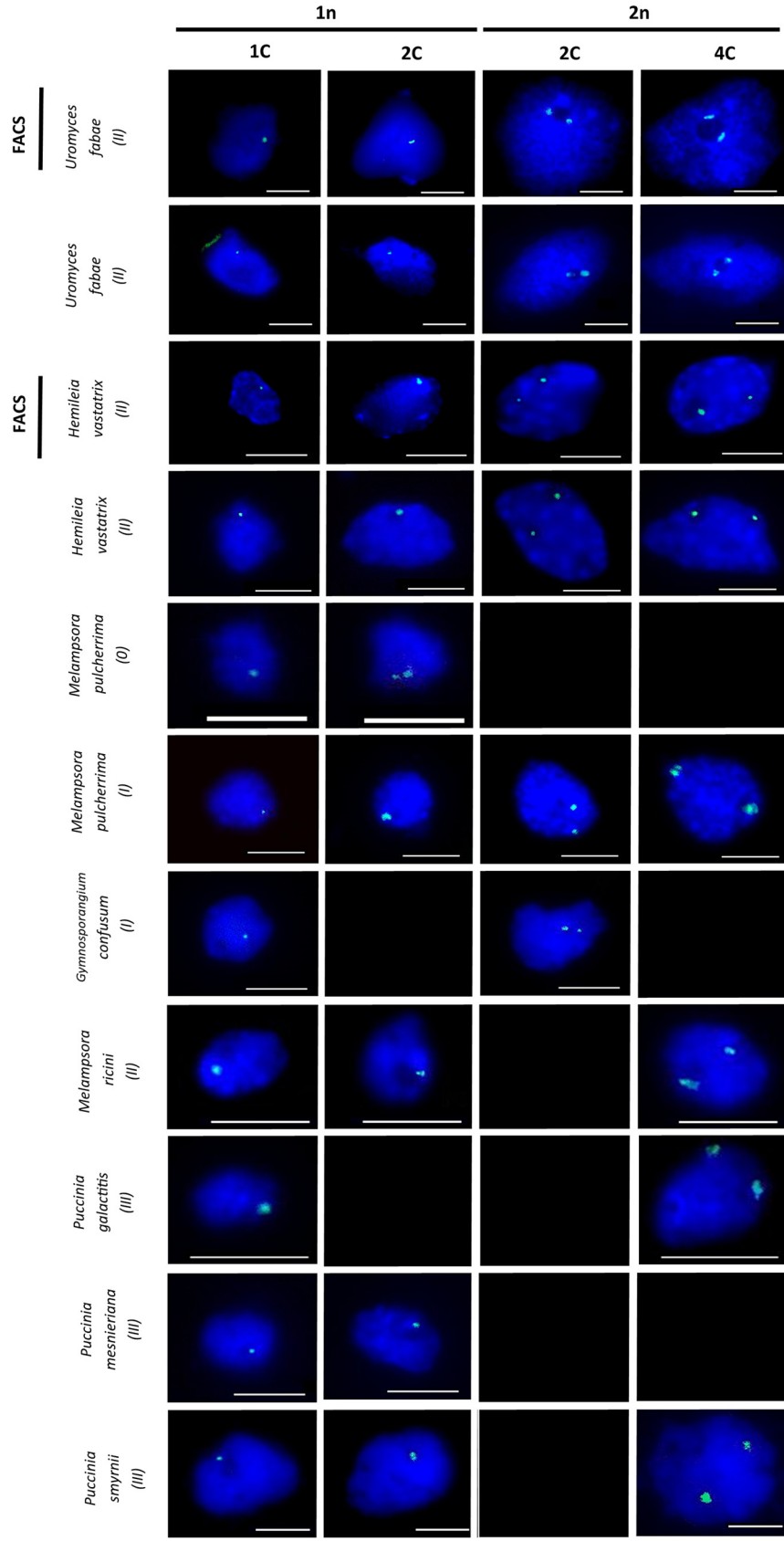

**FIG 5** Fluorescence In Situ hybridization (FISH) with18S and 25S rDNA probes (green detection) in nuclei from different rust species and in diverse cycle stages (indicated in the vertical axis; see Table 1 legend for life cycle stages designation) released by chopping or isolated by FACS according to the ploidy level horizontal. Bar 5 $\mu$m. Black boxes denote no detection.

**TABLE 3** Proportion of haploid nuclei (1n1C, 1n2C), diploid nuclei (2n2C and 2n4C) and its respective percentages in the rust fungi species analyzed

| Species | 1n1C | 1n2C | 2n2C | 2n4C | Total |
|---|---|---|---|---|---|
| *Hemileia vastatrix* | 22 (36%) | 17 (28%) | 14 (23%) | 8 (13%) | 61 |
| *Uromyces fabae* | 29 (37%) | 21 (27%) | 19 (24%) | 10 (13%) | 79 |

urediniospore formation, maturation, germination and appressoria differentiation (10). Our FCM examination of the uredinial infection cycle of *H. vastatrix* (but also of other rust fungi, such as the fava bean rust fungus *Uromyces fabae* and the common bean rust fungus *U. appendiculatus*) could corroborate those findings and are compatible with this hypothesis. However, that work was rapidly refuted (11), and only more recently were those findings reconsidered by FCM analyses (12), although the latter study did not identify the 4C population of nuclei, unlike in our analysis, and no attempts were made to explain such purported cryptosexual process (13). We have further shown that 1C, 2C, and 4C populations cooccur in *Puccinia smyrnii* and *Gymnosporangium confusum* aecial stages. These findings challenge the 'meiosis' hypothesis, as it would therefore mean that a sexual cycle would be occurring both in the uredinial and in the aecial stages, but they are in agreement with ancient reports of two types of nuclei precisely from aecidial, uredinial, and telial stages of several rust fungi (14). The exponential growth in available genome sequences enabled genome evolution studies showing that polyploidy, aneuploidy, accessory chromosomes, and mobile elements activity play an important role in genome evolution, population dynamics and pathogenic specialization (15). However, investigation on the haploidization/diploidization cycles detected in the Pucciniales is still missing.

Over 80 years ago, Savile (14) was impressed by the remarkable variation in size and appearance in nuclei throughout rust fungi life cycles and by the dearth of convincing division figures elsewhere than in the basidium, among the many papers dealing with rust fungi cytology already published by then. He reported two types of nuclei: "unexpanded" nuclei occurring throughout the life cycle of rust fungi, typically associated with haustoria, and also on monokaryotic cells (pycnial stage); "expanded" nuclei occurring in all stages but the pycnial one, including in urediniospores, which subsequently give rise to "unexpanded" nuclei upon germination and appressoria differentiation (as observed in *U. fabae*). These results could correspond to the observations reported in the present work, as FISH analyses do not suggest that diploid nuclei are found in the pycnial stage and 1C nuclei reappear upon urediniospore germination. In a thorough review of the structure and behavior of fungus nuclei, Olive (16) refutes Savile's nuclei types (14) but recognizes substantial variation in nuclear size along rust fungi life cycles, recording the occurrence of very small nuclei in some circumstances while other nuclei are considered the largest in basidiomycetes (namely, in *Coleosporium* and *Gymnosporangium*). Craigie (17), studying *P. helianthi*, reported similar patterns of distribution of "unexpanded" and "expanded" nuclei as those reported by Savile (14), suggesting that their dynamics could be related to the mitotic divisions. Similar differences arose when counting the number of chromosomes in *P. striiformis*, with Wright and Lennard (18) reporting $n = 3$ when analyzing basidiospores while Goddard (19) observed $n = 6$ in germinating urediniospores. Also, the DNA content in nuclei of *P. graminis* f. sp. *tritici* urediniospores was shown to be twice as expected (20), leading the authors to infer that the nuclei were 1n2C. Karyological studies in the Pucciniales became scarcer as other basidiomycetes offered better research approaches (21). Still, in the last major review devoted to rust fungi nuclear behavior, Littlefield and Heath (22) pointed out striking differences in nuclear staining along fungal development, commenting on the scarcity of further work on such phenomenon.

The present work clearly shows that the nuclear cycle of Pucciniales fungi does not obey the general rules in Mycology. In fact, it resurrects ancient scattered findings prompting further research on the genetic and cytological basis of this phenomenon. While remarkable studies into rust fungi life and nuclear cycles were performed over decades based solely on the available microscopic tools, the current breadth of cytological and genomic tools enable genome size estimation, ploidy level determination,

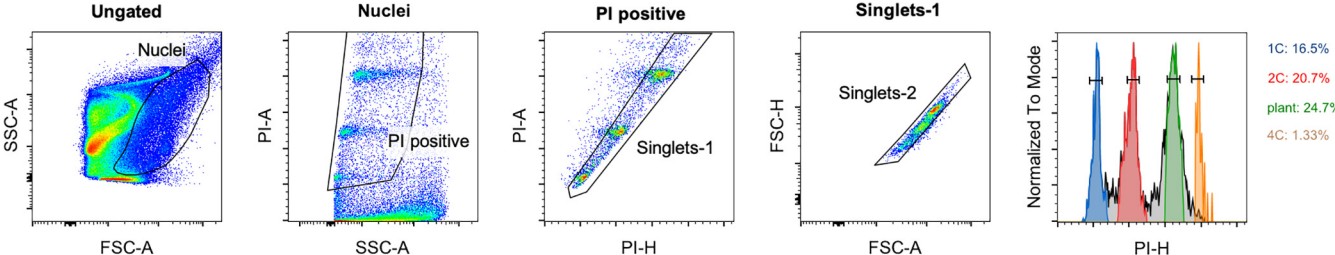

**FIG 6** Example of a gating strategy for identification and FACS sorting of *Hemileia vastatrix* nuclei isolated from *Coffea arabica* infected leaves. PI was detected in the PE channel. Very restricted gates were applied to each nuclei population minimize contamination with neighboring populations and/or debris.

karyotyping and chromosome separation, quantitative FISH, gene expression studies, genome sequencing, among others. Correctly combined with phytopathology, mycology, and histopathology tools, they represent a powerful approach to the elucidation of aspects that were otherwise either speculative or totally elusive.

## MATERIALS AND METHODS

**Collection and handling of fungal samples.** In total, 35 Pucciniales species were analyzed, representing 10 genera from seven families (Table 1). Micro-, demi-, hemi-, and macrocyclic species were selected, and samples were analyzed at the pycnial, aecial, uredinial, and telial stages, as detailed in Table 1, but not all stages were found or were only available in insufficient quantities or mixed with structures from other stages. Four fungi from sister orders to the Pucciniales (class Pucciniomycetes) were also studied, along with another two strains from the class Microbotryomycetes. Additional Basidiomycota and Ascomycota fungi were used for comparison. Rust fungi were collected from nature in Portugal, unless otherwise stated. The remaining fungi were obtained from culture collections. Cultivable fungi were grown on Potato Dextrose Agar (Difco) medium. *Microbotryum lychnidis-dioicae*, *Rhodotorula babjevae* and *Septobasidium carestianum* were analyzed in the yeast phase, while the remaining non-Pucciniales fungi were analyzed in filamentous form. Pucciniales fungi were analyzed on their host plants. The uredinial cycle was monitored in further detail for *Hemileia vastatrix* (isolate CIFC178a on leaves of coffee genotype H147/1), *Uromyces fabae* (isolate UoH-race I2 on 'Inovec' fava bean plants), *U. appendiculatus* (isolate UoH-SWBR on 'Catarino' common bean plants), and *U. striatus* (isolate UoH-Us on a *Medicago arabica* spontaneous plant). Urediniospore germination and appressoria formation were induced *in vitro* on water and on scratched polyethylene film, respectively (23) and *in planta* as described by Loureiro et al. (24).

**Ploidy level and nuclear DNA content determination.** Fungal ploidy level and nuclear DNA content were determined by flow cytometry using the protocols by Tavares et al. (5) for Pucciniales fungi, by Sabatinos and Forsburg (25) for yeasts and by Bourne et al. (26) adapted by Pires et al. (27) for cultivable filamentous fungi. For DNA content estimation, plant and fungal standards used were, respectively, *Solanum lycopersicum* (28), *Raphanus sativus* (28) and *Rhamnus alaternus* (29), and *Inonotus hispidus*, *Colletotrichum acutatum* and *Cenococcum geophilum* (30) (Table 1). Analyses were conducted in a CyFlow Space flow cytometer (Sysmex, Norderstedt, Germany) equipped with a 30 mW green solid-state laser emitting at 532 nm for optimal propidium iodide excitation and emission at 617 nm (532/617 nm), as previously described (31). In most cases, to avoid as much as possible the background noise due to cell wall debris autofluorescence, a gating strategy (following Talhinhas et al. (31)) was employed aiming to maximize the read of positive events and to minimize that of debris, while attempting to maintain consistency among the different samples.

**Isolation of nuclei populations and DNA:DNA fluorescent *in situ* hybridization.** Fungal nuclei populations (1C, 2C and 4C) were collected separately using Fluorescence-Activated Cell Sorting (FACS). Nuclei were subsequently analyzed cytogenetically by Fluorescence *In Situ* Hybridization (FISH) using 35S rDNA probes. This pipeline was subjected to optimization to ensure the most discriminating results. DNA fixation was attempted before and after FACS, using either 70% ethanol, 4% paraformaldehyde, FAA (50% ethanol, 5% acetic acid, 3.7% formaldehyde) or no fixation. Optimal fixation was achieved with 4% paraformaldehyde after FACS.

Fungal (and plant) nuclei were released from the biological material using the Woody Plant Buffer (WPB (32)) as described by Tavares et al. (5) and stained with 50 $\mu$g/mL of propidium iodide (PI; Sigma-Aldrich). Nuclei FACS was performed in a BD FACSAria Ilu instrument (BD Biosciences) according to the strategy depicted in Fig. 6. PI fluorescence was detected out of the blue laser using a 582/42 bandpass filter. The bulk population of fungal nuclei was identified by backgating PI positive events in the forward (FSC) and side (SSC) scatterplot. After the scatter gating, PI-positive nuclei were gated, followed by doublet exclusion using both PE-A/PE-H and FSC-A/FSC-H. Populations of about 1000 particles each of 1C, 2C and 4C nuclei were sorted onto a sucrose drop (100 mM Tris, 50 mM KCl, 2 mM MgCl$_2$, 0.05% Tween 20 and 5% sucrose) in (3-Aminopropyl) triethoxysilane-coated slides. An aqueous 4% formaldehyde solution was placed over the dried sucrose drop for 10 min. After quickly draining the fixative solution, the slides were dehydrated in 70% – 100% ethanol series 5 min each. Dried slides were stored at −20℃ until further analysis.

Fluorescent *in situ* hybridization was performed on FACS isolated nuclei with 35S rDNA probes. These are universal probes and we have isolated the highly conserved 18S and 25S genes from wheat genomic DNA with three pairs of primers: one for 18S gene (18S-FW 5′-ACTGTGAAACTGCGAATGG-3′ and 18S-REV 5′-

CCCGACTGTCCCTGTTAATC-3′); and two for the 25S gene (25S1-FW 5′-CTTAGTAACGGCGAGCGAAC-3′ and 25S1-REV 5′-CACTTGGAGCTCTCGATTCC-3′, 25S2-FW 5′-AACTCACCTGCCGAATCAAC-3′ and 25S2-REV 5′-GCCGAAGCTCCCACTTATC-3′). rDNA gene products were labeled by nick translation with Dig-dUTP (Roche, Switzerland) according to the manufacture instructions and detected with an anti-digoxigenin antibody conjugated to fluorescein isothiocyanate (FITC). DNA:DNA fluorescent *in situ* hybridization was performed as previously described (33, 34). Briefly, after pretreatment of the nuclei with RNase and pepsin, the hybridization mixture comprising formamide, saline sodium citrate buffer, dextran sulfate, sodium dodecyl sulfate, salmon sperm DNA and the labeled probe was applied to the nuclei with a stringency of 74%. The mixture together with the nuclei were denatured at 80°C and were kept overnight after cooling to 37°C. Finally, posthybridization washes were performed with a stringency of 84%. DNA was counterstained with VectaShield Mounting Medium containing DAPI (Vector Laboratories, UK). hybridized slides were analyzed with an epifluorescence microscope Axio Imager.Z1 (Zeiss, Germany). All images were acquired with an AxioVision HRm camera (Carl Zeiss, Germany) using Zeiss filter sets 49 (445/50 nm) and 44 (530/50 nm) for DAPI and FITC, respectively, and were assembled using Adobe Photoshop 6.0.

For some samples, the discrimination from the host plant material and FACS-isolation of rust fungi nuclei was compromised, probably due to inherent and heterogeneous features of the host plant. For this reason, fungal nuclei from various infections stages were also subjected to FISH without previous isolation by FACS. For fungal samples within plant material, $0.5 \times 0.5$ cm pieces of infected leaf material were cut and fixed in 4% formaldehyde (in $1\times$ PBS containing 0.1% Tween 20, pH 7.0) under vacuum for 20 min, rinsed in $1\times$ PBS and chopped with a razor blade in WPB. Nuclei and fungal structures were separated from larger debris using a 100 $\mu$m nylon filter and 20 $\mu$L droplets were dropped over 20 $\mu$L droplets of sucrose solution (5% sucrose, containing 100 mM Tris, 50 mM KCl, 2 mM MgCl$_2$, and 0.05% Tween 20) laying in (3-Aminopropyl) triethoxysilane-coated slides. Slides were air-dried overnight, the nuclei were fixed in 75% ethanol + 25% glacial acetic acid at $-20$°C for 20 min, air dried and used for FISH.

## ACKNOWLEDGMENTS

This research was financially supported by the Fundação para a Ciência e a Tecnologia (Portugal) project PTDC/BIA-MIC/1716/2014, PhD grant 2021.05658.BD (to R.C.) and R&D unit LEAF (UIDB/04129/2020 and UIDP/04129/2020; project EpiRust). Article Processing Charges were financially supported by the 'Veríssimo de Almeida' Plant Pathology Laboratory (ISA/ULisboa, Portugal).

Conceptualization—P.T., S.T., H.A., M.d.C.S., M.M., J.L., L.M.-C. Formal analysis—P.T., R.C., S.T., T.R., M.M., J.L. Funding acquisition—P.T., A.P.R., M.d.C.S., J.L., L.M.-C. Investigation—P.T., R.C., S.T., T.R., H.A., A.P.R., M.M., J.L. Methodology—P.T., S.T., T.R., H.A., M.M., J.L. Project administration—P.T., J.L., L.M.-C. Resources—H.A., A.P.R., M.d.C.S., M.M., J.L., L.M.-C. Supervision—P.T., J.L., L.M.-C. Visualization—P.T., R.C., S.T. Writing—original draft—P.T., R.C. Writing—review and editing—all authors.

We have no conflicts of interest to declare.

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
