## [Reviewer comments · Microbiology Spectrum]

Microbiology Spectrum

Diploid nuclei occur throughout the life cycles of Pucciniales fungi

Pedro Talhinhos, Rita Carvalho, Sílvia Tavares, Teresa Ribeiro, Helena Azinheira, Ana Paula Ramos, Maria do Céu Silva, Marta Monteiro, João Loureiro, and Leonor Morais-Cecílio

Corresponding Author(s): Pedro Talhinhos, Instituto Superior de Agronomia, Universidade de Lisboa

Review Timeline:

Submission Date:	April 11, 2023
Editorial Decision:	April 30, 2023
Revision Received:	May 5, 2023
Accepted:	May 14, 2023

Editor: Michael Klutstein

Reviewer(s): The reviewers have opted to remain anonymous.

Transaction Report:

DOI: <https://doi.org/10.1128/spectrum.01532-23>

April 30, 2023

Prof. Pedro Talhinhos
Instituto Superior de Agronomia, Universidade de Lisboa
LEAF
Tapada da Ajuda
Lisboa 1349-017
Portugal

Re: Spectrum01532-23 (Diploid nuclei occur throughout the life cycles of Pucciniales fungi)

Dear Prof. Pedro Talhinhos:

please refer to the reviewers' comments very carefully, especially to reviewer no. 2 who was more critical of your work.

Link Not Available

Sincerely,

Michael Klutstein

Journals Department
Reviewer comments:

Reviewer #2 (Comments for the Author):

The manuscript "Diploid nuclei occur throughout the life cycles of Pucciniales fungi" describes the analysis of Pucciniales fungi nuclei through cytogenetic and FISH analysis. This study is important in understanding the Pucciniales species and why they are so different to other closely related fungal species. However, it is unclear how the methodologies were benchmarked and validated and more detail in the materials and methods is required. Referring to methodologies from other papers without specifically providing information with what was done is not adequate.

This reviewer is not convinced that genome sizes can be determined from cytometry, and some sort of validation is required via

genome sequencing or other methodologies.

It also seems assumptions are made about nuclear DNA content and signal, as well as signal from the FISH. I would like to see other validations for these experiments or at least these limitations to be addressed and discussed.

It is unclear why certain lifecycles for certain species are chosen, and need to be discussed.

The gating strategy for cytometry looks like for some fungal species cut off the population right down the middle. Why is this? No data given to back this up.

Multiple species are analysed and mean data is shown in several tables - this should include standard errors/deviations.

Reviewer #3 (Comments for the Author):

The paper by Talhinhas et al explores ploidy diversity in multiple species of Puccinales and close relatives.

The main findings of the paper are that Puccinales have a unique nuclear biology, whereby haplontic, diplontic nor haplodiplontic states can co-exist in the same cell.

The results are exciting, and compelling. I also feel can possibly apply to other fungi that have unconventional cellular biology.

The paper is very well written, and I have one minor comment that relates to the speculation that Puccinales represent the only MAJOR EXCEPTION that challenges text-book pre-conceptions.

It would be interesting for the authors to perhaps slightly expand their views outside the Dikarya and see if similar patterns exist elsewhere in the Mycota?

For example, the statement that "karyogamy is immediately followed by meiosis, and thus diploid nuclei should occur only in rust fungi basidia textbook assumptions, is also challenged by recent work on Arbuscular Mycorrhizal Heterokaryons.

Indeed, Sperschneider et al 2023 (BiorXiv) have reported that strains containing TWO parental haplotypes physically separate among thousands of nuclei originated from two parental strains but HAVE YET TO UNDERGO meiosis; a situation (at least partially) similar to that reported here.

This is just one example worth mentioning, but it's possible that some literature search may find more; further increasing the impact of this work.

Staff Comments:

Preparing Revision Guidelines

For complete guidelines on revision requirements, please see the journal Submission and Review Process requirements at <https://journals.asm.org/journal/Spectrum/submission-review-process>. **Submissions of a paper that does not conform to**

Microbiology Spectrum guidelines will delay acceptance of your manuscript. "

Please return the manuscript within 60 days; if you cannot complete the modification within this time period, please contact me. If you do not wish to modify the manuscript and prefer to submit it to another journal, please notify me of your decision immediately so that the manuscript may be formally withdrawn from consideration by Microbiology Spectrum.

Dear Dr. Michael Klutstein

We would like to thank you for considering our work for possible publication in Microbiology Spectrum and would also like to acknowledge the constructive comments raised by reviewers. Bellow are point-by-point responses to those questions, marked in red for clarity. DOIs for all references cited in our responses are given in the end of this letter. Changes to the manuscript file are also highlighted in red.

We hope these modifications and responses are found satisfactory and that you can consider the manuscript suitable for publication.

Best wishes
Pedro Talhinhos

Reviewer comments:

Reviewer #2 (Comments for the Author):

The manuscript "Diploid nuclei occur throughout the life cycles of Pucciniales fungi" describes the analysis of Picciniales fungi nuclei through cytogenetic and FISH analysis. This study is important in understanding the Picciniales species and why they are so different to other closely related fungal species. However, it is unclear how the methodologies were benchmarked and validated and more detail in the materials and methods is required. Referring to methodologies from other papers without specifically providing information with what was done is not adequate.

R: It was our intention to validate the results obtained by FCM, and we have chosen FISH for this. Because n2C samples are indistinguishable from 2nC using FCM, FISH was used to distinguish both populations. A clear correlation between singlet or doublet FISH signals with the nonreplicated or replicated loci, respectively, has been established a long time ago (Selig et al. 1992; Kitsberg et al. 1993; Boggs & Chinault 1997; Singh et al. 2003; Dutta et al. 2009; please refer to the end of this response letter for the DOIs of these references), and has been accepted and used since then. Moreover, the simultaneous use of FCM sorted nuclei and further validation of stages of cell-cycle by FISH is also a common technique (for example Lavoie et al. 2004; Huang et al. 2009) to assess G0/G1 or G2 populations. Furthermore, FISH is a standard technique to assess nuclear ploidy level since the number of individual signals is directly correlated with the ploidy level. In this way, this is a technique used in national laboratories worldwide to diagnose chromosomes abnormalities that can cause disease.

We have added to material and methods section information about the FISH methodology.

This reviewer is not convinced that genome sizes can be determined from cytometry, and some sort of validation is required via genome sequencing or other methodologies.

R: Flow cytometry is the standard technique for genome size measurement in eukaryotes. This has been demonstrated and reviewed exhaustively (e.g., Galbraith 1990; Doležel & Bartoš 2005; Doležel et al. 2007; Bennett & Leitch 2011; Loureiro et al 2021; Sliwinska et al. 2022; and references therein). In spite of the small genome size of fungi, FCM has also been validated and methods optimised for the measurement of fungal nuclear DNA content (e.g., D'Hondt et al. 2011; Bleichrodt & Read 2019; Talhinhos et al. 2021; Čertnerová 2022). Genome size measurement by FCM can in fact be employed to measure the quality and the completeness of genome sequencing initiatives (as an example, see Figure 2 and the discussion of the article by Hill et al. 2021, but also Pflug et al. 2020 and Wyngaard et al. 2022 for examples from animals). The main focus of the present work, however, was not the measurement of fungal genomes, but the

analysis of ploidy level in rust nuclei (still, some fungal genome sizes were newly determined in this study). Flow cytometry of nuclear particles is also a standard technique for ploidy and nuclear cycle analyses (Galbraith et al 1983; DeLaat et al. 1987; Schutte et al. 1995; Johnston et al. 2005; Suda & Travnicek 2006; Ramsey & Ramsey 2014; Fomicheva & Domblides 2023). Therefore, it is clear that the techniques used in this study are solidly validated for the purposes for which they were employed.

It also seems assumptions are made about nuclear DNA content and signal, as well as signal from the FISH. I would like to see other validations for these experiments or at least these limitations to be addressed and discussed.

R: Please see the answer about methodologies.

It is unclear why certain lifecycles for certain species are chosen, and need to be discussed.

R: We have analysed as many rust species as possible. Similarly, we have analysed as many lifecycle stages as possible. Not all lifecycle stages were found for all species. For instance, we have analysed the aecial stage of *Gymnosporangium confusum* on *Crataegus monogyna*, but we were unable to find the uredinial/telial stage on *Juniperus* spp., in spite of multiple surveys. We were also not able to analyse the pycnial stage of this fungus in spite of several attempts, as it turn out to be very ephemeral, yield very limited amount of biomass (insufficient for analysis) and very soon giving rise to the aecial structures. We did attempt performing detailed dissection of in planta fungal structures using Laser Capture Microdissection, but this yielded no reliable FCM results. A clarification on the sampling strategy was added to the Materials and Methods section.

The gating strategy for cytometry looks like for some fungal species cut off the population right down the middle. Why is this? No data given to back this up.

R: In most cases, to avoid as much as possible the background noise due to cell wall debris autofluorescence, a gating strategy (following Talhinhos et al. 2021) was employed aiming to maximize the read of positive events and to minimize that of debris, while attempting to maintain consistency among the different samples. This information was added to the text.

Multiple species are analysed and mean data is shown in several tables - this should include standard errors/deviations.

R: Standard deviation values were included next to average values corresponding to multiple species (Table 2). The main text was also adjusted to encompass the interpretation of these values.

Reviewer #3 (Comments for the Author):

The paper by Talhinhos et al explores ploidy diversity in multiple species of Puccinales and close relatives. The main findings of the paper are that Puccinales have a unique nuclear biology, whereby haplontic, diplontic nor haplodiplontic states can co-exist in the same cell. The results are exciting, and compelling. I also feel can possibly apply to other fungi that have unconventional cellular biology. The paper is very well writtes, and I have one minor comments that relates to the speculation that Puccinales represent the only MAJOR EXCEPTION that challenges text-book pre-conceptions. It would be interesting for the authors to perhaps slightly expand their views outside the Dikarya and see if similar patterns exist elsewhere in the Mycota? For example, the statement that "karyogamy is immediately followed by meiosis, and thus

diploid nuclei should occur only in rust fungi basidia textbook assumptions, is also challenged by recent work on Arbuscular Mycorrhizal Heterokaryons. Indeed, Sperschneider et al 2023 (BiorXiv) have reported that strains containing TWO parental haplotypes physically separate among thousands of nuclei originated from two parental strains but HAVE YET TO UNDERGO meiosis; a situation (at least partially) similar to that reported here. This is just one example worth mentioning, but it's possible that some literature search may find more; further increasing the impact of this work.

R: Thank you for your enthusiasm regarding our results. The results described by Sperschneider et al. (2023) are compelling and we have included a reference to that study in our discussion in order to expand the interpretation of the results outside the Dikarya, as suggested by the reviewer. Still, we would like to keep our discussion mainly centred in the Pucciniales and their sister taxa, so we have not expanded much on AMF. We believe that the strength of our results rely on the analysis of a phylogenetically diverse set of organisms, for all of which the phenomenon is documented, and on the fact that we were able to show that the nuclear ploidy level varies along the life cycle of these fungi.

References used in this response letter

Bennett & Leitch 2011 <https://doi.org/10.1093/aob/mcq258>
Bleichrodt & Read 2019 <https://doi.org/10.1016/j.fbr.2018.06.001>
Boggs & Chinault 1997 <https://doi.org/10.1126/science.220.4601.1049>
Čertnerová 2022 <https://doi.org/10.1002/cyto.a.24485>
D'Hondt et al. 2011 <https://doi.org/10.1111/j.1364-3703.2011.00711.x>
DeLaat et al. 1987 <https://doi.org/10.1111/j.1439-0523.1987.tb01186.x>
Doležel & Bartoš 2005 <https://doi.org/10.1093/aob/mci005>
Doležel et al. 2007 <https://doi.org/10.1038/nprot.2007.310>
Dutta et al. 2009 <https://doi.org/10.1371/journal.pone.0004970>
Fomicheva & Domblides 2023 <https://doi.org/10.3390/mps6010018>
Galbraith 1990 [https://doi.org/10.1016/S0091-679X\(08\)60553-1](https://doi.org/10.1016/S0091-679X(08)60553-1)
Galbraith et al 1983 <https://doi.org/10.1126/science.220.4601.1049>
Hill et al. 2021 <https://doi.org/10.1016/j.fbr.2021.03.003>
Huang et al. 2009 <https://doi.org/10.1007/s00709-009-0051-x>
Johnston et al. 2005 <https://doi.org/10.1093/aob/mci016>
Kitsberg et al. 1993 <https://doi.org/10.1038/364459a0>
Lavoie et al. 2004 <https://doi.org/10.1101/gad.1150404>
Loureiro et al 2021 <https://doi.org/10.1002/cyto.a.24331>
Pflug et al. 2020 <https://doi.org/10.1534/g3.120.401028>
Ramsey & Ramsey 2014 <https://doi.org/10.1098/rstb.2013.0352>
Schutte et al. 1995 <https://doi.org/10.1002/cyto.990060106>
Selig et al. 1992 <https://doi.org/10.1002/j.1460-2075.1992.tb05162.x>
Singh et al. 2003 <https://doi.org/10.1038/ng1102>
Sliwiska et al. 2022 <https://doi.org/10.1002/cyto.a.24499>
Sperschneider et al. 2023 <https://doi.org/10.1101/2023.01.15.524138>
Suda & Travnicek 2006 <https://doi.org/10.1002/cyto.a.20253>
Talhinhas et al. 2021 <https://doi.org/10.1002/cyto.a.24335>
Wyngaard et al. 2022 <https://doi.org/10.1038/s41598-022-10585-2>

May 14, 2023

Prof. Pedro Talhinhos
Instituto Superior de Agronomia, Universidade de Lisboa
LEAF
Tapada da Ajuda
Lisboa 1349-017
Portugal

Re: Spectrum01532-23R1 (Diploid nuclei occur throughout the life cycles of Pucciniales fungi)

Dear Prof. Pedro Talhinhos:

congratulations! the reviewers were satisfied with your revision, and your manuscript is accepted.

Your manuscript has been accepted, and I am forwarding it to the ASM Journals Department for publication. You will be notified when your proofs are ready to be viewed.

Sincerely,

Michael Klutstein
Editor, Microbiology Spectrum
